# A Review of the Effect of Plasticizers on the Physical and Mechanical Properties of Alginate-Based Films

**DOI:** 10.3390/molecules28186637

**Published:** 2023-09-15

**Authors:** Zahra Eslami, Saïd Elkoun, Mathieu Robert, Kokou Adjallé

**Affiliations:** 1Center for Innovation in Technological Ecodesign (CITE), University of Sherbrooke, Sherbrooke, QC J1K 2R1, Canada; zahra.eslami@usherbrooke.ca (Z.E.); mathieu.robert2@usherbrooke.ca (M.R.); 2Research Center for High Performance Polymer and Composite Systems (CREPEC), Montreal, QC H3A 0C3, Canada; 3Environmental Biotechnology Laboratory, Eau Terre Environnement Research Centre, Institut National de la Recherche Scientifique (INRS), Quebec, QC G1K 9A9, Canada; kokou.adjalle@inrs.ca

**Keywords:** alginate, hydrophilic and hydrophobic plasticizers, plasticization mechanisms, films, physical and mechanical properties, water vapor permeability

## Abstract

In recent years, there has been a growing attempt to manipulate various properties of biodegradable materials to use them as alternatives to their synthetic plastic counterparts. Alginate is a polysaccharide extracted from seaweed or soil bacteria that is considered one of the most promising materials for numerous applications. However, alginate potential for various applications is relatively limited due to brittleness, poor mechanical properties, scaling-up difficulties, and high water vapor permeability (WVP). Choosing an appropriate plasticizer can alleviate the situation by providing higher flexibility, workability, processability, and in some cases, higher hydrophobicity. This review paper discusses the main results and developments regarding the effects of various plasticizers on the properties of alginate-based films during the last decades. The plasticizers used for plasticizing alginate were classified into different categories, and their behavior under different concentrations and conditions was studied. Moreover, the drawback effects of plasticizers on the mechanical properties and WVP of the films are discussed. Finally, the role of plasticizers in the improved processing of alginate and the lack of knowledge on some aspects of plasticized alginate films is clarified, and accordingly, some recommendations for more classical studies of the plasticized alginate films in the future are offered.

## 1. Introduction

Over the last century, the massive growth in plastic materials usage has raised environmental concerns [1]. Plastics made from petrochemical-based polymers are highly valorized because of their relatively low cost and good mechanical and barrier properties. They are easy to process and can be used in various applications [2]. Among their applications, almost one-third of their use is for packaging films. However, despite the advantages, petroleum-based waste resulting from packaging or other sources has created many environmental and economic problems. Plastic waste will remain in the environment for a long time (over 50 years), and will not degrade by itself. This can create a highly dangerous situation in which the environment, human health, and marine life are threatened by air and water pollution and soil contamination. Although recycling has been an efficient technique for re-using plastic materials, it is not cost-effective nor applicable to all plastic products [3]. Additionally, dangerous substances are possibly migrating into food from synthetic packaging materials, which is highly threatening to human health [4]. Accordingly, materials capable, in the long term, of replacing plastic with a bio-sourced alternative could be highly promising.

In recent years, research on renewable and biodegradable materials has been widely carried out aiming to substitute traditional fossil-derived polymers with biopolymers. Biopolymers are derived from natural sources, such as polysaccharides, proteins, and lipids [5]. Biopolymers are completely biodegradable and can degrade in the environment over a considerably short time [1]. Among polysaccharides, alginate has attracted commercial attention as a result of its low cost, biodegradability, biocompatibility, low oxygen permeability, non-toxicity, high nutritional value, renewability, and abundance [6]. However, alginate-based films possess poor mechanical properties as well as high water vapor permeability [7], and improving their properties to make them comparable with their synthetic counterparts is a very challenging task.

Due to alginate-based films’ high stiffness and low flexibility, introducing an appropriate plasticizer into the formulation is vitally important to reduce the strong intermolecular interactions between alginate–alginate chains [8]. Accordingly, we need to have detailed knowledge about the influence of plasticizers on the various properties of alginate-based films to be able to develop customized materials. To the best of our knowledge, reviews in this regard are very rare in the literature. Some reviews on the effect of plasticizers are about improving the flexibility of the most common biopolymers such as PLA [9] and starch [10,11], but there is no dedicated article focusing the effect of plasticizers on alginate-based films, in particular. The objective of this review paper is to investigate the effects of different types of plasticizers on water vapor permeability, and the thermal, mechanical, and physical properties of alginate-based films. Therefore, in the first part, alginate’s structure, properties, and applications will be examined, as well as the microstructure of the plasticizers, polymer–plasticizers interactions, and plasticization mechanisms. Moreover, the plasticizers will be classified into two classes comprising hydrophilic and hydrophobic plasticizers and their effects on the plasticized alginate-based films will be studied in detail. Finally, scaling-up challenges and the lack of knowledge about different aspects of plasticized-alginate films will be addressed.

## 2. Alginate: Structure

Alginate is a type of polysaccharide and an anionic linear biopolymer extracted either from brown seaweed or produced from soil bacteria. In brown seaweed, it is present in the cell wall of algae and functions to increase strength and flexibility. Alginate can be present in two forms: the acidic form which is called alginic acid, and the salt form named sodium alginate [7].

From the structural point of view, alginate is a complex mixture of two monomeric units, β-D-mannuronic acid (M) and α-L-guluronic acid (G), and they are joined together by glycosidic (1–4) bonds (Figure 1) [12]. In Figure 1a, the arrangement of the alginate molecules is presented. Three regions of M-blocks, G-blocks, and alternating sequences of MG-blocks can be present in the alginate structure (Figure 1b). As seen in the figure, M and G blocks have different shapes. M block is formed from β (1–4) linkages, so it is a relatively straight polymer, like a flat ribbon with a linear and flexible conformation. However, the G block is formed from α (1–4) linkages, so the resulting chain is buckled, introducing a steric hindrance around the carboxyl groups. This buckled shape provides folded and rigid structural conformations, responsible for the considerable rigidity of the molecular chains of the G block [12,13].

In each type of alginate, the blocks may be distributed in different ways, and the sequence of the bocks seems to be random. In each type of alginate, the blocks may be distributed in different ways, and the sequence of the bocks seems to be random. Different factors such as biological species, geographic origin, the state of maturation, and harvesting time can be determinants in the distribution and sequences of the blocks [15,16]. As a result, the composition, relative proportions of uronic acids (M/G ratio), and molecular weight may vary from one type of alginate to another. However, according to the literature, the M/G ratio can be altered on a laboratory scale by converting mannuronic acid residues into guluronic acid residues in the polymer chain or vice versa, so it is possible to manipulate alginate’s intrinsic properties [12].

It has been proven that the physical properties of alginates depend heavily on the M/G ratio. Accordingly, for each industrial application, a certain type of alginate with a certain M/G ratio can be used. For example, in some applications, higher strength and stiffness are needed; hence, low M/G ratio alginate would be a better choice. Conversely, for applications in which more flexibility and elasticity are needed, alginate with a higher M/G ratio should be utilized [17]. Accordingly, it is highly important to be able to measure the relative proportions of M and G by using different methods [12]. Nuclear magnetic resonance (NMR) spectroscopy and circular dichroism analysis (CD) are two methods that have been used for measuring this ratio. For example, Avella et al. [17] used NMR to measure the percentage of the guluronic and mannuronic acids content in alginate, as well as the average number of consecutive guluronate moieties in G-block structures. The composition of the M/G ratio of alginate was also analyzed by Feng et al. [18] according to CD spectra.

## 3. Alginate: Properties and Application

Alginate is a kind of biomaterial that has found many applications in different industries. Alginate is a gelatinous material with thickening, gel-forming, and film-forming characteristics; hence it is used extensively in paper coating, textile printing, wastewater treatment (separation membranes) [19], cosmetic and food industries (food additives and food packaging), and in pharmaceutical and medical applications (drug delivery, wound dressing, tissue engineering).

There are three main properties associated with alginate, making it a very popular biopolymer to be used in different applications. Firstly, it can thicken aqueous solutions. When dissolved in water, alginate can increase the viscosity of the solution. For example, in the canning process, alginate can act as a thickening agent and increase the overall viscosity of the food. The second property of alginate that is very important in the food industry, as well as the medical industry, is its ability to form a heat-stable gel in the presence of calcium ions or other divalent cations [20]. The divalent cations can be trapped between guluronic acid residues to form a stable, 3D structure generally called the “egg-box” structure [17]. In contrast to other polysaccharides such as agar, or carrageenan, there is no heat required for the dissolution and formation of the gel, which is beneficial when aiming to avoid any detrimental effects of heat on the material. The third property of alginate is the ability to form a film or coating [20]. The possibility of crosslinking alginate-based films through zipping guluronate (but not mannuronate) chains together in an “egg-box” conformation makes alginic acid an interesting biopolymer for film development, allowing it to be used in drug and food preservative applications [7].

In various fields of biomedicine and the pharmaceutical industry, alginates have drawn special attention in the past few years. They are frequently used as an encapsulation of drugs, proteins, and cells [21], as substances for cell culture, in drug delivery systems and tissue engineering, and as binders for tablets [22]. The development of thin alginate films is highly important for pharmaceutical formulation. For instance, alginate films and coatings are used extensively for covering tablets and pellets to not only mask the unpleasant odor and taste of the drug but also to protect the drug from oxidation and hydrolysis [7]. In addition, alginate can effectively be used as a wound dressing where it acts as a drug delivery system to promote wound healing as a result of its hydrophilicity and high water sorption ability. These systems can maintain water and encased bioactive substances and drugs such as antibiotics and then release them gradually in a controlled manner [7,23].

Wastewater treatment is another attractive application of alginate-based materials. The presence of heavy metal ions in water resulting from industrial activities has created many environmental problems for not only humans but also for aquatic ecosystems because of their serious toxic effects. Biopolymers such as alginate have the potential to uptake metal ions from solutions because of the high amounts of free hydroxyl and carboxyl groups located on the backbone of the alginate structure [24]. Alginate-based absorbents have been investigated for removal of several metal ions, such as Pb^2+,^ Hg^2+^, Cd^2+^, Cu^2+^, Ni^2+^, Co^2+^, Mn^2+^, and Cr^3+^, from water [25]. In addition, it was demonstrated that the alginate films show higher performance in removing Pb^2+^ from a solution compared with alginate beads because of the higher exposed area of the films [26].

Alginate is a tasteless, odorless, and transparent component, which makes it an ideal candidate for food packaging applications in which it can be utilized in different forms and shapes such as films [27] (monolayer or multilayer films and composites), and coatings [15]. These films and coatings made from alginate or other bio sources can be edible or nonedible. In active food packaging production, edible films and coatings have been used to a great extent [28]. Among active food packaging systems, oxygen, and moisture scavengers are the largest segments, accounting for 37% and 16% of the global market of active food packaging, respectively [7].

Edible films and coatings have a very strong potential for use in food packaging production. They have been studied widely for food preservation since their microstructure allows them to act as a barrier and preserve food through stabilizing and localizing the activity and controlling the release of food preservatives (antimicrobials, antioxidants) at the interface. However, high water solubility and the poor mechanical properties of alginate-based films and other biopolymers such as starch, agar, chitin, and chitosan have limited their applications [13]. Accordingly, there has been a great effort in recent years to improve the properties of biopolymer-based films [13,15,17,29,30,31]. Nowadays, many works are attempting to decrease the film permeability of alginate-based films to water vapor [31,32,33,34]. Alginate is a hydrophilic and water-soluble material, and although it possesses good oxygen barrier properties, its water vapor permeability (WVP) is high as a result of being hydrophilic. Alginate can show swelling behavior when exposed to water, which can accelerate the water uptake and the WVP [15,34]. Low WVP is especially important when using alginate in food packaging applications to prevent food spoilage. This problem can be addressed by incorporating divalent cations such as calcium as a crosslinking agent, causing the alginate films to become water-insoluble and have lower water vapor permeability [32]. Another problem associated with pure alginate is its brittleness and poor mechanical properties, especially after crosslinking [34]. While alginate films have a high Young Modulus (YM), usually their Tensile Strength (TS) and Elongation at Break (EB) are low. To improve these properties, plasticizers are mostly added to increase the flexibility of the film by decreasing intermolecular forces among polymer chains [13]. For food packaging applications, mechanical properties are of vital importance, so it is highly important to have general knowledge about the effect of plasticizers on mechanical properties, as well as on physical, thermal, and barrier properties, to be able to develop successful packaging materials. In the following section, plasticizers and their effects on alginate-based film will be discussed in more detail.

## 4. Plasticizers

In general, plasticizers can be defined as low molecular weight (between 300 and 600) [35], high boiling point materials which are added to a film-forming polymer to enhance its flexibility, durability, processability, and workability [36]. Their addition can avoid shrinking during storage [29], and in some cases reduce costs [37]. Additionally, the incorporation of plasticizers into the polymer can affect some physical properties of the polymer such as the viscosity, degree of crystallinity, glass transition temperature (T_g_), hardness, and density [15,35].

### 4.1. Plasticization Mechanisms

The plasticization mechanism can be explained by several theories. During the 1940s, the “lubricity theory” and the “gel theory” were developed. In these theories, scientists attempted to explain critical phenomena observed in plasticized polymers [38]. According to the lubricity theory, plasticizers reduce intermolecular friction between the polymer chains. Increasing the flexibility of the plastic part leads to the polymer molecules slipping over each other. Accordingly, plasticizers lubricate the movement of the polymer molecules by reducing their resistance to sliding. However, in the gel theory, polymers are considered as three-dimensional honeycomb structures in which the resistance of these 3D structures rather than the internal frictions (as the lubricity theory states) is primarily responsible for the rigidity of the polymers. In this theory, the plasticizer reduces the number of attachment points from chain to chain, letting the polymer be deformed without breaking by reducing the aggregation of the polymer molecules.

Later, during the 1950s, some other models were suggested; however, among them, the free volume theory gives more precise explanations of the plasticizing effect [37,39,40]. This theory attempts to explain the glass transition temperature (T_g_) reduction which occurs with increasing plasticizer content. This theory originated when scientists attempted to explain various concepts related to materials such as thermal expansion coefficients, specific volume, viscosity, and the relationships between these properties and some variables related to polymer structure, such as molecular weight. This is why the free volume theory is still used for explanation of the plasticization mechanism. Although different authors have contributed to developing this theory, Fox and Flory were the first to postulate the idea [41].

#### Free Volume Theory

The internal space between polymer chains is called the free volume. In rigid polymers, the molecules are very compact, so there is very little free volume between the polymer’s chains. The main function of plasticizers is to enhance the free volume within the polymer by decreasing the intermolecular forces between chains, increasing chain mobility, and making the polymer rubbery and flexible [40,42]. Since the increase in free volume leads to increased motion of polymer molecules, to study the plasticization effect it is vitally important to study different ways of increasing free volume.

There are some models correlating free volume fraction to the glass transition temperature for polymers. For example, according to the empirical Boyer–Simha rule, the free volume fraction (*f_exp_*_)_ is related to the cubic expansion coefficient (*β*) of liquid (*L*), and glass-like (*G*) polymers as well as the glass transition temperature (*T_g_*) [38]. The fractional free volume of a liquid (*f*) is defined as *f* = v_1_/v_0_, where v_1_ is the free volume and v_0_ is the occupied volume.
(1)$$f_{exp}\approx (\beta_{L}-\beta_{G})T_{g}\approx 0.11\pm 0.02$$

Moreover, according to the Williams–Landel–Ferry (WLF) approach, the fractional free volume and temperature have a linear correlation [43].
(2)$$f=f_{0}+\beta_{f}\left(T-T_{0}\right)$$

According to this theory, the free volume of a glassy polymer at T_g_ is 2.5% of the total volume. The WLF equation is one of the most successful theories of free volume.

Principally, the free volume in the polymer structure comes from three main motions: the motion of chain ends, side chains, and the main chain. These motions, and therefore, the free volume of a polymer can be increased by different methods. Firstly, the increase in the number of end chains achieved by decreasing the molecular weight of the polymer can promote the free volume between chains. Secondly, both internal (increasing the number or length of side chains) and external plasticization (incorporation of a lower molecular weight material that acts as a plasticizer) can induce this effect, and lastly, elevating the temperature also can increase the internal space of the polymer.

In Figure 2, the schematic illustration of the effect of the most common plasticizer, glycerol, on alginate chains is presented. As can be seen in the figure, glycerol is small on a molecular basis, containing three OH groups. Its low molecular weight allows it to penetrate into alginate chains, and increase the free volume of the polymer by creating secondary bonds with polymer chains and filling up the empty spaces. This can prevent intermolecular interactions of polymer chains, making the polymer more flexible.

To experimentally measure the T_g_ of polymers, DMA (dynamical-mechanical analysis) is commonly used to determine the parameter known as tan δ. The value of tan δ provides information about changes associated with the movement of the polymer chains and their viscoelastic behavior [3]. Generally, in the tan δ profile of alginate-based films, two main transitions can be identified. There is a transition around 100 °C due to the glass transition (α-transition) and another at low temperatures (β-transition) related to the motion of hydroxyl groups connected to water molecules [44,45]. For example, Chen et al. [44] found that the T_g_ of the polymer decreased from 110 °C for the unplasticized polymer to about 60 °C for the glycerol-plasticized alginate film. Avella et al. [17] also evaluated the changes in the T_g_ of alginate by the incorporation of glycerol as a plasticizer. They added 33, 45, and 50%wt glycerol to the polymer. According to the DMA results, the addition of 33%wt glycerol did not affect the T_g_ of the polymer; however, introducing 45 and 50%wt glycerol led to a 30 °C decrease in T_g_ from 120 °C for the neat alginate to 90 °C for the plasticized polymer.

### 4.2. Classification of the Plasticizers

The chemical structure of the chosen plasticizer is highly important since the degree of plasticity of the biopolymer is heavily dependent on the plasticizer type. Accordingly, the plasticizer’s chemical composition, the presence of functional groups, and molecular weight should be taken into consideration. Moreover, when selecting a plasticizer for polymeric systems, different criteria such as compatibility between the polymer and plasticizer; the concentration of plasticizer; processing requirements; physical, thermal, and mechanical properties of the final product; cost-effectiveness; and toxicity should be considered [35,42].

The molecular weight of the plasticizer is a very important parameter. Plasticizers with lower molecular weight can easily diffuse into polymer chains and change the physicochemical properties of the polymer [36,46]. It is well known that water is very small on a molecular basis, and that is part of the reason it is the most effective plasticizer. Water is added widely to polymers as a plasticizer. It is especially added to thermoplastics and elastomers to avoid temperature rises during the hot-compounding process. However, water is highly volatile and can evaporate at low temperatures, so other plasticizers such as polyols are usually added to polymers together with water [19,47]. However, the main problem associated with low molecular weight plasticizers is their tendency to migrate in the long term. Migration is the transfer of the plasticizer from the polymer (rich phase) to other materials (poor phase) that are in contact with the polymer. The transfer occurs through diffusion, and the recipient phase can be water, air, the surface of the polymer, or any other materials in contact with the polymer. The molecular size, shape, and chemical nature of the plasticizer, temperature, and structure of the recipient material can affect the rate of migration of the plasticizer from the polymer [48]. For example, glycerol can migrate to the surface of the polysaccharide or the contacted material. Glycerol creates a secondary bond with the polymer, and its small size can allow it to diffuse to the surface of the polymer or leach to the water. As a result of this plasticizer loss, the polymer becomes brittle with poor mechanical properties. On the other hand, hydrophilic plasticizers such as glycerol can also diminish the barrier properties of the films as they contain hydroxyl groups capable of creating a hydrogen bond with water and increasing the hydrophilicity of the plasticized film. This increased hydrophilicity is responsible for the increased water vapor permeability of the films, which is undesirable for packaging applications [36].

Plasticizers and polymers can interact in two different ways: externally and internally. In external plasticization, although there are physical interactions between plasticizer and polymeric chains, plasticizers do not attach chemically to the polymer by primary bonds (they attach to the polymer by hydrogen bonding—a type of dipole–dipole attraction between molecules), and as a result, they can be lost by migration or evaporation [35]. In contrast, in internal plasticization, plasticizers react with the biopolymer through co-polymerization or grafting and become a part of the polymer chain, so surface migration will be hindered [49]. Nevertheless, external plasticizers are used widely in biopolymers, simply because they are easy to apply.

Among the abovementioned criteria, compatibility between the polymer and plasticizer is of major importance. If the polymer and plasticizer are not compatible, phase separation occurs and the final properties of the product deteriorate. Different parameters such as polarity, dielectric constant, hydrogen bonding, and hydrophilicity can be good indicators for deciding whether plasticizers and polymers are compatible or not [35]. Among those parameters, the water solubility of plasticizers is highly important. Accordingly, plasticizers can be categorized into two major groups: water-soluble (hydrophilic) and water-insoluble (hydrophobic).

Although alginate is a hydrophilic biopolymer, both hydrophilic and hydrophobic plasticizers have been used in various studies to increase its flexibility. In Table 1 the most important studies on plasticized alginate films and their most prominent results and findings are tabulated. The type of plasticizer can strongly affect the barrier properties, water uptake, and mechanical properties of the biopolymer. Generally, water-soluble plasticizers such as glycerol, sorbitol, and polyethylene glycol increase WVP and water uptake of the polymer by increasing the moisture-sensitivity of the polymer, but hydrophobic plasticizers such as triacetin, triethyl citrate, fatty acids [50], and vegetable oils usually improve barrier properties of the polymer by blocking the voids and creating a tortuous pattern in the structure [33]. However, the water-insoluble plasticizers need to be dispersed well in the aqueous solution to prevent phase separation and deterioration of the mechanical properties of the polymer [51]. The effects of water-soluble and water-insoluble plasticizers on various properties of alginate-based films will be discussed in more detail in the following sections.

#### 4.2.1. Water-Soluble Plasticizers

Water

Water can be considered a natural, low-molecular-weight but volatile plasticizer for most biopolymers. The water content of biopolymers readily changes with the environment’s relative humidity through sorption/desorption phenomena [22]. Water molecules can markedly reduce the glass transition temperature of the biopolymer by increasing the free volume between the polymer chains. Moreover, it is the main solvent of alginate, and there is a strong chemical interaction between alginate and water. As a result, water can be considered the most powerful natural plasticizer [40,52]. Treenate et al. [54] investigated the effect of water as a co-plasticizer (glycerol/water or sorbitol/water) on the T_g_ of chitosan/alginate films plasticized with 25, 40, and 50% *w/w* plasticizer. The DMA results showed that there was a dramatic decrease in the T_g_ of the plasticized films, especially for the film plasticized with glycerol. They claimed that the increase in water content led to the solvating of the plasticizer molecules in the water; as a result, the plasticizing ability of those plasticizers was enhanced. Accordingly, a more flexible film could be obtained by using water as a co-plasticizer [54].

For a better understanding of water permeation through edible films when water is either in its vapor or its liquid state, Hambleton et al. [79] studied the physicochemical properties of films based on iota-carrageenan and sodium alginate at three different relative humilities, 0%, 43%, and 84%. According to the Schroeder paradox, for the same activity differential, the mass transfer of permeate depends on its physical state. Water as a permeate plays a key role in the diffusion mechanism for most food systems, and it can modify the matrix structure and induce the diffusion of hydrophilic molecules into the polymer. The water vapor transfer rate (WVTR) results, in their study, showed that increasing the humidity differential led to increased WVTR. As water acts as a plasticizer in hydrophilic films, it can decrease the density or local viscosity by promoting the diffusion of molecules. This has been shown by other authors who indicated that the increase in relative humidity led to the increased permeability of the polysaccharide-based films due to swelling or plasticization [31,51]. Nevertheless, the liquid water transfer rate (LWTR) showed a completely different behavior than WVTR for both biopolymers, as the Schroeder paradox suggests. Since the transfer equilibrium was reached very quickly, Hambleton et al. concluded that the LWTR of both films was several orders of magnitude higher than the WVTR because there was no transfer resistance caused by the stagnant layer as it was supposed in WVTR. However, they suggested that Iota-carrageenan-based films had a different swelling behavior compared with sodium alginate-based films. Alginate-based films swelled very quickly, before penetrating the liquid water inside the films. The cross-linked chains of the alginate-based films known as the ‘‘egg-box’’ structure limited the available sites for water absorption and promoted the partial solubility of film components.

In the study performed by Olivas and Barbosa-Cánovas [31], the effect of RH on WVP (water vapor permeability) of alginate-based films was studied. According to their results, the WVP of films investigated under a RH differential of 100–0% was higher than that of films conditioned at 76–0% RH. This indicates that the WVP of films can be affected by the water activity of products as well as the RH of the environment. The decrease in the capacity of films to function as barriers to water vapor and gases with increasing RH has been reported in other studies [80,81], in which it was concluded that water may play the role of plasticizer in hydrophilic polymers such as alginate.

The plasticizing efficiency of water on the mechanical, thermal, and physical properties of alginate has been investigated widely in other studies [50,53,55]. Barbut and Harper [55] studied the effect of different relative humidity on the mechanical and physical properties of “dried” alginate films. According to their results, the 57% RH film was less transparent than its 100% RH counterpart because of the presence of salt crystals on the film conditioned at 57% RH. In addition, films conditioned at 57% and 100% RH had different mechanical properties: all of the films conditioned at 100% RH had higher elongation at break and lower tensile strength and Young’s modulus than their corresponding films conditioned at 57% RH, and the alginate films with glycerol had remarkably lower tensile strength when conditioned at 100% compared with those conditioned at 57% RH. They concluded that these differences in mechanical properties of films conditioned at 57% and 100% RH can be attributed to the plasticizing effect of water in the films. The mechanical properties of alginate-based films at different RH were also investigated by Hambleton et al. [79]. They showed that the elastic modulus and tensile strength decreased with increasing relative humidity, but the relative humidity affected elongation at break oppositely. This means that water has a plasticizing effect on alginate. According to the DSC results, increasing the moisture content led to a decreased glass transition of the plasticizer-enriched phase, which confirms the plasticizing effect of water on the biopolymer. Olivas and Barbosa-Cánovas [31] proved that the mechanical properties of alginate-based films are strongly affected by RH. By increasing RH, there was a decrease in tensile strength and an increase in the elongation of all films, indicating the plasticizing efficiency of water. In addition, when adding a plasticizer, the mechanical properties of films were more affected by changes in RH.

The effect of water as a plasticizer in “wet” alginate films has been investigated in a few studies [55,56,82,83]. Although “wet” films have been used commercially, there are few published scientific studies about them, and most of the papers have investigated the properties of ‘dried’ biopolymer films to be used after casting. “Wet” films are films that contain approximately 90% to 95% water, and are used mostly in the sausage industry. For the preparation of these films, the co-extrusion method has been used in recent years, and the films are not dried before being applied to the product. Co-extruded alginate casings are a type of edible biopolymer film that is mostly used for sausage casings [56]. Harper et al. [83] studied the mechanical properties of “wet” alginate-based films. According to the results, the elongation at break values was much higher than the values reported in the literature for “dried” alginate-based films. They concluded that the water in the ‘wet’ films acts as a plasticizing agent since plasticizers are known to increase the elongation at the break of films.

In addition, water can also be used as a “destructuring” agent in the thermo-mechanical mixing method to reduce the intermolecular bonds of the polymer and reach a molten state and homogenous phase [19,47,57]. To decrease the processing temperature, water and other plasticizers are added because polysaccharides such as alginate usually melt after degradation temperature, and this can lead to degraded polymeric material with diminished properties [47]. Gao et al. studied the effects of water as a destructuring agent and glycerol as a nonvolatile plasticizer on the thermal and mechanical properties of thermo-mechanically produced alginate films [19]. They showed that under the thermo-mechanical mixing method, the alginate particles were significantly destructured as demonstrated by SEM and XRD, and the mechanical properties. According to the XRD results, the unplasticized alginate film showed sharper peaks at 2Ɵ = 13.5° and 21.6° compared with the neat alginate powder. The reason might be the rearrangement of polymer chains and macromolecular networks when adding water during the film-forming process, which can improve the mobility of alginate chains. They also measured the water content of the alginate-based films after equilibration at 25 °C and 57% relative humidity since water also acts as a plasticizer in the system and then affects the thermal and mechanical properties of samples. According to the results, the water content of the films increased remarkably with increasing plasticizer (glycerol) concentration because glycerol has three hydroxyl groups that can bond to water molecules through hydrogen bonding and increase the water content of the films. In addition, the water content in plasticized alginate was higher than that of starch and chitosan-based film [84,85], indicating the fact that alginate possesses higher hydrophilic characteristics compared with other polysaccharides. However, according to the results obtained by Jost et al. [15], moisture content decreased with the addition of plasticizers such as glycerol and sorbitol because of their lower water-binding capability and less hydrophilic nature compared with alginate. The plasticizing efficiency and hygroscopic effect of glycerol have been reported widely in various studies and will be discussed in further detail in the following sections.

In another study on the effects of two plasticizers (glycerol and sorbitol) and their mixtures on alginate properties, the effects of water as both a plasticizer and destructuring agent in the preparation of alginate-based films by thermo-mechanical mixing method were highlighted [47]. It was reported that the water content of the glycerol-plasticized film was higher than that of the sorbitol-plasticized film because of the higher hydrophilicity of glycerol. Since water can also act as a plasticizer, the glycerol-plasticized films had higher plasticity than their sorbitol-plasticized counterparts. However, when the glycerol concentration exceeded 40 wt.%, the glycerol-plasticized alginate showed lower elongation at break than the sorbitol-plasticized alginate. This behavior has been reported in other studies on starch-based film [86] and is due to a segregation phenomenon caused by glycerol accumulation in different parts of the sample. This phenomenon has been also seen when high amounts of water (high relative humidity) are incorporated in other polysaccharides like starch, even with the addition of a low amount of glycerol [87]. These findings indicate that a high amount of water in the system can destroy the strong hydrogen bonds between starch–glycerol and starch–starch species, instead, the weaker hydrogen bonds between starch–water and glycerol–water species have been created, resulting in a decreased elongation at the break of the films.

Souza et al. [57] prepared CS (corn starch) and CS–SA (corn starch–sodium alginate) films by using single- and twin-screw extruders and a hot press, with 15% glycerol and different amounts of water. The mechanical properties of CS–SA blends were affected by the SA composition and water content. To control the water content in the blends, the samples were conditioned at different levels of relative humidity (RH). At low RH, 5%, the films were fragile and difficult to test. For all samples, the increase in water content to 9 and 15% led to a change in the state of the films from glassy to rubbery materials, a decrease in Young’s modulus and tensile strength, and an increase in the elongation at the break of the samples. Hence, they concluded that the macromolecules comprising the system can form a stronger entangled network at higher water contents.

However, water has a permanent equilibrium with the environment and is volatile at low temperatures. As a result, nonvolatile plasticizers are also added to the polymer to increase the flexibility of the final product. It is worthwhile mentioning that more stable properties can be obtained by the incorporation of nonvolatile plasticizers since the evaporation and migration of the plasticizer to the surface of the product can be prevented significantly during storage [19].

2.Polyols

Polyols such as glycerol, ethylene glycol, diethylene glycol, tri-ethylene glycol, polyethylene glycol, xylitol, sorbitol, and mannitol are considered effective plasticizers to improve the properties of biopolymer films. In the case of alginate-based films, it has been reported that those plasticizers can improve the microstructure, ductility, and flexibility of the resulting plasticized alginate films [13,17].

Glycerol

Glycerol is a triol (a polyol with three hydroxyl groups) that is highly soluble in water. It can be recovered from the trans-esterification of fat and oils in biodiesel plants as well as by saponification and hydrolysis reactions in oleo-chemical plants. Glycerol is a by-product of those reactions, and it may contain impurities such as water, soap, remaining catalysts, salt, and free fatty acids [88]. It is extensively used in either pure or mixed form in the food industry as a food ingredient, as well as in cosmetic products and pharmaceutical formulations, thanks to its superior properties and low toxicity. In addition, glycerol is commonly used as a plasticizer for edible films since it has been approved by the FDA as a safe food additive [40]. Glycerol is a polar and nonvolatile material (boiling point of 290 °C) with high availability [77]. These characteristics combined with the acceptable mechanical and thermal properties of final glycerol-plasticized biopolymer films make glycerol an effective and one of the most widely used plasticizers in the industry. Glycerol has three hydroxyl groups which allow the creation of strong hydrogen bonds with the hydrophilic polymer and increase intermolecular spacing by reducing internal hydrogen bonding. The addition of glycerol reduces the brittle nature of alginate and provides the desired extent of flexibility [13,15,17,55]. As a result, glycerol-plasticized alginate has been extensively used for making composites or films [1].

In the study performed by Avella et al. [17], the influence of increasing amounts of glycerol (33, 45, and 50%) on the chemical and physical properties of sodium alginate-based films was investigated. They analyzed two types of alginates: one of them was high in guluronic acid content with a higher molecular weight (with the identification code “Ap”) and the other was low in guluronic acid content with a lower molecular weight (with the identification code “Ar”). The DSC results for the plasticized Ap samples showed that the incorporation of 33% glycerol decreased the water-releasing temperature (endothermic peak) from 130 to 110 °C, but the increase in glycerol content to 45 and 50% led to an increased water-releasing temperature. Because of the interaction between alginate and glycerol, water is expelled from the alginate structure, leading to a lower water-releasing temperature. However, for the samples with higher amounts of glycerol, there was an excess amount of glycerol that interacted with water, and as a result, the release of the bound water occurred at a higher temperature. The analysis of the DSC curves of the plasticized Ar system showed that the endothermic peak of samples containing 33 and 45% glycerol increased compared with the neat alginate, but there was a decrease in the endothermic peak of samples containing 50% glycerol, indicating a clear plasticizing effect of glycerol in the blends containing high amounts of glycerol. This means that the introduction of glycerol in the first two blends (33% and 45% of plasticizer) did not appreciably change the molecular network of the polymer; however, only in a blend containing 50% glycerol could an early plasticizing effect be seen, which was confirmed by the DMTA results, indicating there was a lower glass transition temperature of the glycerol in a blend containing 50% glycerol compared with the other samples. To interpret these results, they suggested that the higher molecular weight of the Ap system is an indicator of longer average chain length and the higher fraction of guluronic acid is responsible for a buckled structure of the polymer, which results in the plasticizer being entrapped quite firmly in the structure. Therefore, the addition of only 33% of glycerol caused a clear plasticizing effect on the polymer. In contrast, the lower molecular weight of the Ar polymer means a shorter average chain length of the polymer, and the higher M/G ratio corresponds to a higher fraction of mannuronic acid in the Ar polymer. As a result, the Ar polymer was more flexible than the Ap system, and water could not be strongly retained in the structure and was released at lower temperatures compared with Ap polymer, according to the DSC results.

Considering the mechanical properties, the Young’s modulus of both plasticized systems was lower than that of the neat polymer. In the Ap polymer, noticeable drops in the elastic modulus and stress at the break of the blends plasticized with 45 and 50% glycerol were detected. It was suggested that the excessive amount of glycerol is responsible for this behavior since it changes the chemical interactions among the polymer, glycerol, and water. This kind of behavior is also highlighted in other studies, confirming that the increase in glycerol content beyond a certain limit can lower the film strength [13,47,58]. In the Ar system, there were slight decreases in the modulus and stress at the break of blends containing 33 and 45% plasticizer in comparison with the neat polymer, and only in the blend with 50% glycerol was a considerable plasticizing effect observed.

In the work performed by Giz et al. [13], the synergistic effects of various concentrations of glycerol (0–30%) and calcium chloride (0 to 2%) on the thicknesses, mechanical and thermal properties, transmittance, water vapor permeability, and swelling properties of alginate-based films were investigated. The analysis of the stress–strain behavior of the films showed that the fracture strain values of all plasticized films increased with the incorporation of glycerol since it provides more mobility in polymer structure by replacing hydrogen bonds. Nevertheless, calcium chloride incorporation decreased the fracture strain but increased the tensile strength of the films. Generally, with high amounts of calcium chloride (1 or >1%), only small amounts of glycerol lead to an increase in tensile strength but larger amounts of glycerol have a deteriorating effect. They suggested that the increase in either glycerol or calcium chloride beyond certain limits leads to a weaker film. The investigation of the effects of various concentrations of calcium chloride and glycerol showed that their effects on the mechanical properties are nonlinear and synergic. It was also shown that crosslinking with calcium chloride had a negligible effect on tensile behavior without glycerol, but the incorporation of glycerol resulted in cross-linked films with better mechanical properties. By optimizing the mechanical properties of the samples containing various amounts of glycerol as a plasticizer and calcium chloride as a crosslinking agent, they concluded that films with 10% glycerol had the best mechanical properties. However, they suggested that wide parametric studies are needed to accurately predict the properties. In addition, the swelling behavior of the films was investigated by performing the test in three solutions: water, acetic acid, and citric acid. It was shown that swelling of the films decreased in all three solvents with increasing calcium chloride concentration, but swelling increased with increasing glycerol content in the water and citric acid.

Gao et al. [19] investigated the effects of glycerol as a plasticizer on the thermal and mechanical properties of alginate-based films. The produced films were transparent, flexible, and homogeneous, but the surface of the films containing more than 30 wt% glycerol was sticky, indicating that glycerol migrates to the surface of the film at higher contents. In addition, they reported that with rising plasticizer content, the film absorbed more water, since glycerol contains three hydroxyl groups which react with water molecules through hydrogen bonding and increase water content in the system. Olivas and Barbosa-Canovas [31] reported the same behavior in glycerol-plasticized alginate with increasing glycerol content, showing that the films plasticized with glycerol exhibited higher moisture content than pure alginate, especially at higher RH. In addition, the thermal stability of plasticized and pure alginate was assessed by thermogravimetric analysis (TGA) in two different environments (helium and air). It was found that the degradation temperature of the plasticized films was lower than that of neat alginate, indicating that the addition of glycerol reduced the thermal stability of alginate-based films, which is in agreement with the work on plasticized chitosan [89]. Moreover, the mechanical properties of the films were investigated using a uniaxial tensile testing machine. The stress–strain curves of the films showed that without a plasticizer, the material was brittle, but the incorporation of 10 and 20 wt% glycerol made the films more ductile, with plastic deformation. However, with high amounts of plasticizer (more than 30%), the modulus of the films reduced considerably with a decreased elongation at break as a result of the segregation phenomenon. Such behaviors were also reported for plasticized starch [90] and amylose amylopectin films plasticized with 30 wt% of glycerol [91].

Although glycerol is considered a nonvolatile plasticizer, it can evaporate even at low temperatures (about 10% loss at 25 °C) [77]. The large drying surface area in the solvent casting method is mainly responsible for this plasticizer loss. However, there are very few research works investigating the dry matter of glycerol in the film after drying. Bagheri et al. [77] found that the films dried at higher temperatures contained lower amounts of glycerol because of evaporation loss. Because of this plasticizer loss during drying at higher temperatures, the amount of film-forming solution decreased and the produced films possessed lower thickness compared with the films dried at lower temperatures. Silva et al. [59] also obtained the same result regarding the thickness of films. They determined the glycerol content in alginate film after drying at 30–60 °C and reported that at temperatures above 40 °C, a significant amount of glycerol was lost during drying. Moreover, the films dried at 60 °C were less plasticized due to glycerol loss during drying at higher temperatures.

Ethylene glycol

Ethylene glycol is an important organic material that is used extensively in industrial processes. It has unique properties and applications and has been synthesized through different methods such as reaction processes derived from fossil fuels, as well as biomass-based resources. Ethylene glycol is the simplest diol with a molecular structure that contains two hydroxyl groups [92]. Ethylene glycol can change the physical, thermal, and mechanical properties, as well as the water vapor permeability of polysaccharide-based films such as starch and alginate. Ethylene glycol, diethylene glycol, tri-ethylene glycol, and polyethylene glycol with various molecular weights are commonly utilized for plasticizing films.

In the study by Pongjanyakul and Puttipipatkhachorn, the effects of glycerin and polyethylene glycol 400 (PEG400) as plasticizers on the physicochemical properties of sodium alginate magnesium aluminum silicate (SA-MAS) micro-composite films were investigated [30]. The objective of their work was to use SA-MAS as a coating material for modifying drug release from tablets such as acetaminophen (ACT) and study the effects of plasticizers on the efficiency of the coatings. According to the findings, both plasticizers (glycerin or PEG400) could be used to improve the physicochemical properties of the SA-MAS films to control the drug release from tablets. As shown by FTIR spectroscopy, there is a possibility that the plasticizers interact with alginate and MAS through hydrogen bonding. Moreover, powder X-ray diffraction (PXRD) results revealed the possibility of intercalating glycerin and PEG400 into the silicate layers of MAS. Compared with glycerin-plasticized film, the film with PEG400 had higher crystallinity, which was confirmed by the higher intensity of the peak in the PXRD graph of the PEG400-plasticized film. This can lead to different thermal behavior of the films with PEG400, which was investigated by DSC. It was shown that glycerin and PEG400 had different thermal behavior than the neat alginate. The intensity of the exothermic and the endothermic peaks decreased with increasing glycerin or PEG400 content, which could be related to the recrystallization and phase transition of the films after heating. In addition, the degradation temperature (exothermic peak) of the films decreased by 34 °C with the incorporation of 50% glycerin, but only by 4 °C with the incorporation of 50% PEG400, indicating a different crystal structure of the films plasticized with PEG400, as confirmed by PXRD results. Considering the mechanical properties, glycerin was a better plasticizer than PEG400 and gave more flexibility to the films. This is in agreement with another study on the effects of both plasticizers on alginate films [93]. The main reason for the better plasticizing efficiency of glycerin is that it is a smaller molecule than PEG400. In addition, the higher crystallinity of PEG400-plasticized film increased the rigidity of the film compared with that with glycerin. The results for water vapor permeability (WVP) revealed that both plasticizers decreased the WVP of alginate when the concentration was 10–30%. They explained that intermolecular hydrogen bonds were formed by penetrating the plasticizers between alginate chains and silicate layers of MAS. This can increase the tortuosity of the pore channels in the films and reduce the WVP of the film. However, the increase in plasticizer amounts by up to 50% led to an increase in the WVP of the films because of the hygroscopic nature of both plasticizers. In addition, the films plasticized with PEG400 had a lower WVP than that of the films with glycerin, because of the higher crystallinity of the films with PEG400, as shown by the PXRD results. The lower WVP of PEG-plasticized films was also highlighted in other studies. For example, El Miri et al. [42] investigated the effects of various polyol-based plasticizers, such as glycerol, diethylene glycol (DEG), and polyethylene glycol (PEG), on the properties of alginate-based films filled with cellulose nanocrystals (CNC), and concluded that the WVP of films plasticized with PEG and DEG was lower than that of GLY-plasticized film. They concluded that this can be explained by the degree of hygroscopicity and the chemical structure of different plasticizers. Although GLY contains three hydroxyl groups, PEG and DEG have one and two hydroxyl groups, respectively, in their structures. As a result, GLY can hinder intermolecular and intramolecular bonding in the network, increasing the free volume and penetration of water molecules into the film and increasing the WVP of the film, but this phenomenon did not occur in the films plasticized by PEG and DEG, thus the WVP was lower. The lower viscosity of PEG- and DEG-plasticized alginate compared with glycerol-plasticized solutions can be explained similarly since the glycerol molecule with three OH groups suppressed the polymer chain motion and increased the viscosity of the film-forming solution. They also reported that PEG- and DEG-plasticized films had better mechanical properties than glycerol-plasticized films. Because of the lower molecular weight of glycerol compared with the two other plasticizers, the glycerol-plasticized film was more flexible. However, it had a lower tensile strength and Young’s modulus than the PEG- and DEG-plasticized films.

Olivas and Barbosa-Cánovas (2008) [31] developed alginate-based films with two different M/G ratios (0.45 and 1.5) with the incorporation at a 40% level of four different plasticizers: PEG-8000, glycerol, fructose, or sorbitol. It was observed that after drying, all the plasticized films were transparent except the film plasticized with PEG-8000. The PEG-plasticized film was opaque and white, which could be related to the phase separation between alginate and PEG. In other studies on edible films plasticized with PEG, the white residue on the surface of the film was observed at a high molecular weight of PEG (higher than 1450) [94,95]. Olivas and Barbosa-Cánovas analyzed the films by moisture sorption isotherms, and all plasticized films, including the PEG-8000-plasticized film, adsorbed more water in comparison with the non-plasticized films. However, the results obtained from mechanical testing showed that the incorporation of PEG-8000 diminished both tensile strength and elongation at the break of the alginate-based films in contrast with glycerol-plasticized film. It is also worth noting that although all plasticized films including PEG showed similar sorption isotherm curves, even at high RH, PEG did not increase the elongation at the break of the film. No difference between the elongation of PEG-8000-plasticized and unplasticized films was recorded, probably because of the phase separation between alginate and PEG, as indicated by the white surface of the PEG-plasticized film.

Sugar alcohols

Sugar alcohols are another group of polyols that are produced from the reduction of sugars. These polyols can be produced from the hydrogenation of three main sources: monosaccharides, disaccharides, and a mixture of saccharides and polysaccharides. For example, sorbitol (glucitol) is a polyol with six carbons that is used extensively for plasticizing polysaccharide-based films. It is produced by the hydrogenation of glucose (a kind of monosaccharide with six carbons) [96]. Apart from sorbitol, mannitol, maltitol, and xylitol [60,61] are examples of sugar alcohols used for plasticizing alginate.

Jost et al., (2014) [15] investigated the effect of two plasticizers (glycerol and sorbitol) on the microstructure, WVP, and mechanical properties of alginate-based films. The concentrations of glycerol and sorbitol were set at 20–40%, and 30–50%, respectively, because of sufficient flexibility at these ranges. Color measurement of the samples showed that the plasticized films were more transparent than pure alginate. Moreover, the sorbitol-plasticized films were more transparent than the glycerol-plasticized films since sorbitol molecules have a better steric fit to the alginate structure than glycerol molecules. SEM micrographs also showed that sorbitol-plasticized films were smoother than both neat alginate and glycerol-plasticized films. Considering the mechanical properties, both plasticizers showed a plasticizing effect on alginate-based film by decreasing the tensile strength and increasing the elongation at break. However, glycerol seemed to be more effective since the incorporation of 30% glycerol was as effective as the incorporation of 50% sorbitol. They explained that because the molecular weight of glycerol is almost half that of sorbitol, the amount of glycerol needed is more than half the amount of sorbitol needed to achieve the same mechanical properties. This shows that although glycerol is a more effective plasticizer based on mass content, the plasticizing efficiency of sorbitol was higher at the molecular basis. However, in another study, Tong et al. [36] concluded that glycerol had the greatest plasticization effect on pullulan–alginate–carboxymethylcellulose blend films compared with three other plasticizers (sorbitol, xylitol, and fructose) because lower amounts of glycerol were needed to increase the elongation at break.

In addition, Jost et al. [15] found that water vapor and oxygen permeability of alginate increased with increasing glycerol content, but not with increasing sorbitol. They explained that the different behavior of glycerol and sorbitol could be related to different plasticizing mechanisms. It has been proven by water vapor and oxygen permeability, as well as water content results, that the plasticizing behavior of glycerol can be explained by the free volume theory. However, the sorbitol behavior did not obey this theory but the gel theory. The sorbitol molecules had a good fit in the alginate macromolecule network, which not only reduces the intermolecular interactions between alginate chains but they can also bond chemically to the polymer through six hydroxyl groups. As a result of this interaction between alginate and sorbitol, the flexibility of the film increases and at the same time, water molecules will interact with alginate. Accordingly, the barrier properties can be controlled, even at higher sorbitol concentrations. Lower WVP levels of sorbitol-plasticized films were also highlighted in other studies [31,97,98]. Parris et al. [99] demonstrated that, unlike glycerol, the presence of sorbitol in alginate-based films can decrease the WVP of the films. However, they concluded that this difference is due to the lower efficiency of sorbitol compared with glycerol in reducing intermolecular interaction between alginate chains rather than their different plasticizing mechanisms.

Santana and Kieckbusch [60] investigated the effects of polyols (glycerol (G), xylitol (X), and mannitol (M)) as plasticizers on WVP and the physical and mechanical properties of alginate-based films. They observed that the incorporation of glycerol and xylitol led to more transparent and uniform films with higher hydrophilicity compared with mannitol-plasticized film, which was whitish before crosslinking. According to the SEM results, this whitish appearance is related to crystal formation in the film, which disappeared after immersion in CaCl_2_ solution. The plasticized films were thicker than the neat alginate films, with the M-film being even thicker. They explained that the high thickness of the G- and X-films could be attributed to the increase in the free volume between chains, making the films less dense, but the high thickness of the M-film is highly likely due to the phase separation between alginate and mannitol. However, the mannitol-plasticized film had the lowest WVP among the samples because it is less hygroscopic. Moreover, considering the mechanical properties, they concluded that glycerol and xylitol were more efficient plasticizers than mannitol.

3.Sugars

Monosaccharides, disaccharides, and oligosaccharides are the sugars most commonly used as plasticizers in polysaccharide-based polymers, such as starch and alginate, because of the similarity between the molecular structures of the plasticizers and polymers. Monosaccharides are the simplest carbohydrates, and they cannot be hydrolyzed into smaller carbohydrates. Glucose, fructose, sucrose, and lactose are some examples of sugars that come from different sources such as fruits or milk.

The plasticizing efficiency of fructose was investigated by Olivasa and Barbosa-Canovas [31]. According to their results, because of the similarity in the sorbitol and fructose molecular weights (182.17 and 180.16, respectively), the plasticized films with these plasticizers possessed similar properties. At higher relative humidity (76% and 100%), the WVP of the films containing fructose was the lowest among the plasticizers. They reported that the low WVP of fructose-plasticized alginate films is comparable to that of lipid-plasticized whey protein emulsion film [100]. However, there was no considerable difference between the WVP of fructose and sorbitol-plasticized films at other RHs. They also showed that the films containing fructose and sorbitol have similar tensile strength and elastic modulus. The similar behavior of sorbitol and fructose is also highlighted in another study. According to Tong et al. [36], sorbitol- and fructose-plasticized blend films exhibited the lowest and similar elongation at break values at any given plasticizer concentration compared with glycerol and xylitol-plasticized films, with the fructose-plasticized film being even more brittle with higher tensile strength and lower elongation at break. However, the moisture contents of films containing fructose were lower than those of films containing sorbitol, maybe because of their different molecular structure. As a result of this behavior, the WVP of fructose-plasticized films was less affected by the plasticizer concentration. Zhang and Han [101] also reported that monosaccharide-plasticized pea starch films were more resistant to water vapor permeation than polyol-plasticized films, although their mechanical properties were comparable.

Fruit and vegetable purees have a good potential to be used in making edible films. Since they are high in sugars such as glucose, fructose, and saccharose, they can have a plasticizing effect on the produced film [102]. In the study by Kadzińska et al. [63], the physical properties of apple puree–sodium alginate edible films incorporated with vegetable oils were investigated. They found that in the samples containing apple puree, the glass transition temperature was lower, indicating the plasticizing effect of natural plasticizers such as fructose and glucose present in apple puree.

#### 4.2.2. Water-Insoluble Plasticizers

All the plasticizers introduced above, such as sorbitol, glycerol, and polyethylene glycol, have hydrophilic characteristics. This hydrophilicity can lead to an increase in WVP and a tendency of the polymer to absorb more water from the environment. Hydrophobic plasticizers, such as vegetable oils, triacetin, tributyl citrate, triethyl citrate, and fatty acids are good alternatives to solve this problem.

Oils

Oils are of special interest for use as plasticizers in polysaccharide-based films because they are natural and hydrophobic; in addition, they are safe for food packaging applications since they have low toxicity and a low tendency to migrate to the surface as a result of their high molecular weight. In some research works, oils such as castor oil [64], olive oil [61], soybean oil [65], coconut oil [66], canola oil [67], cinnamon oil [68,69], oregano essential oil [51], and garlic oil [70] have been used as both antibacterial agents and plasticizers in alginate-based films. However, in these research studies, glycerol or other low molecular weight plasticizers were the main plasticizers in most cases, and oils were introduced for controlling the hydrophobicity and barrier properties, as well as bacterial and microbial activity of the films, but they had also a plasticizing effect on the films, usually decreasing tensile strength and increasing elongation at break of the polymer [51,64,65,67,69].

Benavides et al. [51] investigated the effect of oregano essential oil (OEO) on the antibacterial, optical, mechanical, and WVP of an alginate-based film. Before being applied to the film-forming solution, OEO was mixed with Tween 80 to create a uniform and stable dispersion in the alginate matrix. Tween and Span are the most common surfactants used for obtaining a uniform solution in different research works when adding hydrophobic plasticizers, such as oils to hydrophilic polymers like alginate [51,61,65,67,68,69,71]. In some other studies, however, a diluted solution of oil from ethanol was used to decrease the surface tension of coating-forming emulsions [64,66,70,103].

Benavides et al. [51] showed that the incorporation of OEO affected the appearance and physical properties of alginate film, making it less transparent and more flexible, with a lower WVP compared with neat alginate. They reported that elongation at the break of the films increased significantly with the addition of OEO, ranging from 2.2% for neat alginate to 3.7% for the film containing 1.5% of OEO, while the tensile strength of the OEO-alginate films decreased by 56% compared with the control sample. These results were consistent with those obtained by other researchers [64,70,72]. They explained that the incorporation of oils usually reduces the TS of the hydrophilic polymer because oils weaken the films by reducing the intermolecular interaction between polymer chains. On the other hand, at room temperature, oil in the liquid state is in the form of small droplets dispersed through the polymer structure that can be easily deformed, which enhances the film’s flexibility. As a result, oils can act as plasticizers by reducing the TS and increasing the EB of the films. However, in some studies, different results have been obtained. For example, in the study by Frank et al. [68], no significant differences between the tensile strength of the cinnamon essential oil/alginate films and control alginate films were observed, and the EB of cinnamon essential oil/alginate films decreased significantly with increases in the concentration of oil. They suggested that at high essential oil concentrations, stressed regions in the film may be created as a result of the formation of discontinuities between two phases, which can lead to rupture and decreased EB. In another study by Gutierrez-Jara et al. [65] about the effect of soybean oil and degree of crosslinking on the physical and mechanical properties of alginate-based films, it was found that at high calcium chloride concentrations, the EB of alginate decreased with increasing oil concentrations. They explained that the rigid surface layer created by calcium chloride as a crosslinking agent predominated over the plasticizing effect of the oil, making the film even more rigid with low EB.

It has been reported in many research works that the WVP of alginate-based films is generally decreased with the incorporation of oil [51,64,65,67]. It has been suggested that the hydrophobicity of oil is the main reason for this behavior, which makes the film more hydrophobic and as a result, reduces the absorption of water molecules. They also suggested that the lower porosity and higher tortuosity in the polymer microstructure brought by oils are other important reasons for the improvements in the WVP of films. In contrast, some studies reported an increased WVP of films with increasing oil content [69,70] because of the formation of pores via changes in the internal structure of the matrix.

Other water-insoluble plasticizers

Apart from oils, other hydrophobic plasticizers, such as fatty acids and citrates, have also been used for plasticizing alginate-based films.

Fatty acids are part of the structure of fats, oils, and waxes. Fatty acids can be defined as any long chain of hydrocarbon, with a single carboxylic group and aliphatic tail. Fatty acids, such as stearic acid, lauric acid [50], and oleic acid [3], have been used as plasticizers in a few studies. In the study conducted by Chen et al. [50], increases in the opacity of soy protein isolate/sodium alginate films as well as changes in the homogeneous structure of the films with the incorporation of both stearic acid and lauric acid were reported. As a result of the heterogeneous structure, the intermolecular interactions of the matrix might have been weakened, which led to the decreased tensile strength of the films. In addition, the elongation at the break of samples decreased with the addition of fatty acids because of the interaction of the polymer with polar plasticizers rather than fatty acids, which led to the deterioration in the flexibility of the films. However, at higher concentrations of lauric acid, elongation at the break of the film increased considerably since lauric acid has a shorter carbon chain compared with stearic acid, and hence, had a better plasticizing effect due to the higher mobility of the molecules. The water vapor permeability results showed that although the barrier properties of the films improved at lower concentrations, they diminished at higher concentrations. The researchers explained that the excessive amounts of fatty acids created a discontinuity on the film surface, which led to increased WVP values. Azucena Castro-Yobal et al. also reported that by adding oleic acid to alginate film, the WVP value slightly decreased compared with the control sample as a result of the hydrophobic nature of the fatty acid.

Citrate plasticizers are another type of hydrophobic plasticizer whose effect on alginate films has been investigated in some studies [33,34,44,93]. These plasticizers are tri- or tetra-esters that are produced by the esterification of citric acid. Citric acid is obtained from citrus fruits, sugar cane, and beetroots, and it is also used as a plasticizer [40,104]. Sharmin et al. [34] investigated the effect of citric acid (CA) as a crosslinking agent on the physical, mechanical, and barrier properties of alginate film and found that CA can act as a plasticizer at higher concentrations. CA was added at 0.5, 1, and 2 w/v% to alginate. At low concentrations (0.5 and 1 w/v%), the water vapor transition rate (WVTR) of the films decreased, indicating the creation of hydrophobic ester groups between citric acid and alginate chains. However, at higher concentrations, the WVTR increased because of the plasticizing effect of CA at higher amounts. On the other hand, according to mechanical testing, the plasticization effect of CA was seen only at high concentrations, with an increase of 475% in the elongation at a break of the 2% CA-alginate film compared with the control sample. These researchers explained that the presence of unreacted CA in the structure led to reduced intermolecular bonding between alginate chains; hence, CA acts as a plasticizer when added in excess amounts.

In another study, the effects of two hydrophilic (glycerol) and hydrophobic (tri-butyl citrate (TC)) plasticizers and their combination on the physical properties of alginate were investigated [33]. The incorporation of TC caused the films to become opaque, which is related to the fact that the films plasticized with TC or glycerol/TC had a globular and sponge-like microstructure, as confirmed by SEM. These authors also reported that the water solubility and WVP of the films decreased with the addition of TC because of its hydrophobic nature. They also reported that the tensile stress and elongation at the break of the films increased and decreased, respectively, as a result of higher interactions between alginate chains and TC brought by the high polarity of TC. In addition, only one glass transition temperature appeared in the DMA graphs, indicating a homogenous mixture. However, the T_g_ of the films increased with increasing TC concentration. Similar results were observed in some other studies. For example, in the study by Chen et al. [44], unlike other incorporated plasticizers, triacetin did not increase elongation at the break of chitosan/alginate-based films. Instead, the triacetin-plasticized film was brittle, and thermally stable, with increased recrystallization. Remuñán-lópez and Bodmeier [93] also reported that the inclusion of 30% tri-ethyl citrate led to opaque and brittle films.

## 5. Drawback Effects of Plasticizers on Alginate-Based Films

The interactive forces between alginate chains are responsible for the shrinkage of the dried film, making it brittle and fragile. Accordingly, adding a plasticizer to the film-forming solution is essential to increase the flexibility of the final product [33]. However, the incorporation of the plasticizer can lead to poor mechanical properties and low thermal stability. In addition, the vast majority of the literature has reported that the incorporation of hydrophilic plasticizers such as glycerol may increase the water solubility and WVP of the film. These effects are generally expected and accepted as the side effects of plasticizers and have been discussed extensively in the previous sections. Several methods have been recommended to alleviate these problems, including blending with other polymers such as chitosan [105], starch [57,106], or polyvinyl alcohol [107,108]; crosslinking [13,32]; blending with hydrophobic compounds; and adding reinforcing agents such as cellulose nanocrystals (CNC) [42] and layered silicates [30].

However, in some cases, adding plasticizers may lead to some undesirable and unexpected effects. Segregation phenomenon (phase separation), anti-plasticization effects, leaching, migration, and evaporation from the surface are the unwanted problems associated with plasticizers. To avoid the segregation phenomenon and anti-plasticization effects, an optimal amount of plasticizer should be added to the polymer. On the other hand, leaching, migration, and evaporation from the surface of the polymer are usually caused by the incorporation of low-molecular-weight plasticizers such as glycerol, and these problems can be avoided by applying an effective plasticization process [48].

The segregation phenomenon or phase separation can occur at high amounts of plasticizer. According to Gao et al. [47], although the elongation at the break of the film increased with the incorporation of the plasticizers, it decreased when glycerol content exceeded 40%. However, for sorbitol-plasticized films, this reduction in EB did not occur. Jost et al. [15] also reported that although the most suitable amount of glycerol as a plasticizer for alginate is 20–40 wt.%, the optimal concentration for sorbitol is 30–50 wt.%. At high amounts of glycerol, the formation of glycerol-rich micro-phases led to the segregation phenomenon. This phenomenon was observed for other plasticized polysaccharides such as starch at high glycerol contents [86]. It was also reported that the presence of a high amount of water, even at a low concentration of polyol, can have the same effect [47]. These results are consistent with some other results obtained by casting alginate-based films. For instance, according to Silva et al. [29], phase separation with plasticizer exclusion can be observed on the surface of the film when the plasticizer concentration is above a critical limit.

Another unexpected effect of plasticizers is called anti-plasticization. This phenomenon is usually caused by the addition of smaller amounts of plasticizer and can lead to increased tensile stress and T_g_ and decreased elongation at break. This phenomenon can be explained by the fringed micelle theory, as shown in Figure 3. The crystallites in the structure of the polymer can be formed by lining up the long chains compactly, which increases the rigidity of the polymer. On the other hand, amorphous areas have more free volume, and plasticizers tend to situate themselves in these areas. With low amounts of plasticizer, the free volume of the amorphous region increases slightly, and consequently, the number and size of the crystallites increase as a result of a new redistribution of the configurations in the structure. Accordingly, the rigidity of the polymer is promoted by the inclusion of less plasticizer [38,109]. In another study, the incorporation of polyglycerol increased the T_g_ of alginate [45]. It was concluded that polyglycerol had an anti-plasticization effect on alginate because of the presence of the large amount of hydroxyl groups in polyglycerol, which promote the interaction between chains by hydrogen bonding. This can have a hindering effect on the polymer, which reduces the molecular mobility of the system, hence T_g_ increases. In another study, glycerol, sorbitol, xylitol, and fructose used as plasticizers had an anti-plasticization effect on alginate when added below 7% *w/w* dry basis [36]. As a result of this effect, the tensile strength increased significantly compared with unplasticized film, while there was a reduction in elongation at break and moisture content of the polymer. They explained that this effect could be related to the interaction of plasticizer molecules with hydrophilic side groups, which reduce the free volume in the matrix and suppress the chain motion in the polymer. Consequently, tensile stress increased. However, when increasing the plasticizer content from 14% to 25% *w/w*, tensile stress decreased progressively, reversing the anti-plasticization effect to the plasticization effect.

Leaching, migration, and evaporation are other problems associated with plasticizers. Generally, once the plasticizer is added to the polymer, it should be permanently retained in the structure. Losing the plasticizer is undesirable and may change the properties of the system [48]. Water-soluble plasticizers such as glycerol can be lost by leaching when in contact with an aqueous solution [60]. Silva et al. [29] reported that the hygroscopic nature of glycerol is responsible for most of the water uptake. They explained that the leaching of glycerol is the main reason for increases in the water solubility of the film with increasing glycerol content, since after 24 h of exposure to water, the films lost their flexibility. However, although this problem can lead to alginate films with inadequate stability properties, it has received little attention in the literature.

## 6. Other Plasticization Approaches

As a result of the destructive effects of plasticizers and numerous problems created by their incorporation into alginate polymer, some studies have promoted other methods rather than utilizing the low-molecular-weight plasticizers for the plasticization of alginate. Internal plasticization, using other polymers as plasticizers, and various drying conditions are some measures to plasticize alginate-based films.

### 6.1. Internal Plasticization

An internal plasticizer is defined as a chemical group incorporated by chemical reaction to the polymer for plasticization purposes. The main advantage of internal plasticizers is that the plasticizers are firmly attached to the structure and cannot be lost by leaching or migration. However, internally plasticized systems usually have poorer mechanical and thermal properties compared with externally plasticized systems [48].

Graft copolymerization is an effective method for plasticizing alginate. Işıklan et al. [73] prepared graft copolymers of sodium alginate with itaconic acid in an aqueous solution using ceric ammonium nitrate as the redox initiator under an N_2_ atmosphere. Grafting was confirmed using FTIR. As shown by DSC, the T_g_ of the grafted alginate was lower than that of pure alginate. They concluded that this could be related to the plasticizing effect of grafting as an internal plasticizer. Similar observations were also found in another study. Zohuriaan-Mehr et al. [74] modified various natural polysaccharides such as sodium alginate using ceric-initiated graft polymerization of acrylonitrile under an inert atmosphere and found that the grafted chains could act as internal plasticizers and accordingly reduced the T_g_ value.

### 6.2. Using Other Polymers as Plasticizers

Polymers, such as pectin, lignin, carrageenan and starch, have been added to alginate in different studies to not only plasticize alginate by decreasing the intermolecular forces, but also to modify its thermal, mechanical, and barrier properties.

In the study by Harper et al. [56], wet films made from 0.25% LM pectin, iota- and kappa-carrageenan, and modified potato starch in 5% alginate were found to have a significantly greater elongation at break than the control alginate films. This indicates that these carbohydrates can act as a plasticizer in the films by reducing alginate–alginate chain interactions. However, with the incorporation of potato starch, commercial cellulose, LA gellan gum, or extracted cellulose, the elongation of alginate films did not experience any significant change. The tensile strength of all the films was similar, except for the pectin-alginate films, which had a considerably higher TS than the other films. The plasticizing effect of pectin on alginate and an increase in tensile strength at the same time were also reported by Di Donato et al. [75]. In their study, 10% *w/w* recovered lemon and fennel wastes were used as alginate plasticizers to improve their flexibility, and it was found that fennel had better performance in plasticizing alginate. They confirmed that both plasticizers contain pectin-like polymers. There was a decrease in T_g_ and degradation temperature, as well as an increase in elongation at the break of the plasticized films corresponding to plasticization. However, unexpectedly, the tensile strength of the films increased with the incorporation of plasticizers as a result of increased physical entanglements between alginate chains and polar groups of the added materials.

It has been reported in some studies that lignin can act as a plasticizer of gelatin and starch-based systems but only when added in medium concentrations [110,111]. In the study by Aadil et al. [76] on the improvement of the physicochemical properties of alginate by using acacia lignin, it was found that lignin has an apparent plasticizing effect on alginate by reducing the intermolecular interactions between chains and decreasing the tensile strength of the films.

### 6.3. The Effect of Drying Conditions on the Plasticity of Alginate

One of the limiting factors of the solvent casting method as a conventional method for producing alginate-based films is the long drying time at ambient temperature [77]. Increasing the drying temperature and airflow rate during the drying process are measures that can be taken to decrease the drying time. However, the characteristics and physicochemical properties of the films can be significantly influenced by these conditions [112]. Although in some studies the rigidity of films increased [59,77], in others, higher drying temperatures promoted the plasticity of the films [22,78].

Wong et al. [78] investigated the effects of drying temperature (40, 60, and 80 °C) and alginate/solvent ratio (1–4% *w/w*) on the mechanical properties of alginate-based films. According to their thermomechanical analysis, the films prepared at 80 °C were more plasticized than the films produced at lower temperatures. They stated that the properties of films dried in a single stage and at higher temperatures were similar to those of films dried in two stages and at lower temperatures (40 °C). Accordingly, using a single-stage drying process at high temperatures is preferable, since apart from the process being convenient, it can give rise to higher plasticity of the material. On the other hand, Ashikin et al. [22] found that the plasticity of alginate-based films decreased when increasing the drying temperature from 40 °C to 60 °C because of heat-induced alginate chain–chain interactions. However, there was an increase in the plasticity of the films at 80 °C as a result of the formation of air bubbles and the reduction in alginate molecular weight at high temperatures. However, in some studies, high drying temperatures led to a more brittle film. For instance, Silva et al. [59] prepared alginate films under different drying temperatures (30, 40, 50, and 60 °C) and reported that oven-dried films were less flexible than ambient-dried films because of the considerable evaporation of glycerol at high temperatures. Bagheri et al. [77] also obtained the same result with glycerol-plasticized films when increasing the drying temperature from 25 °C to 90 °C. Their findings indicated that all the properties of the films, except color, can be significantly affected by the drying temperature. However, different air flow rates (0–12 L/s) did not change the characteristics of the films considerably. While the elastic modulus and tensile strength increased when applying higher temperatures, elongation at break, water vapor permeability, and drying time of the films decreased. The SEM and AFM results (Figure 4) showed that increasing drying temperature led to a smoother surface with a compact cross-section, which was also confirmed by the XRD results, which showed a higher degree of crystallinity in oven-dried films. However, DTG analysis showed that although the loss peak related to glycerol shifted to a lower temperature, the maximum degradation temperature of alginate did not change. Consequently, it can be inferred that while the plasticity of the films decreased with elevated drying temperature, the overall characteristics of the alginate-based films were improved.

## 7. Discussion

### 7.1. Scaling Up Challenges

For producing alginate-based films at a laboratory scale, the solvent casting method is generally used. In this technique, alginate and other additives, such as plasticizers, are dissolved in deionized water, and then cast in Petri dishes to evaporate the water [15]. In the majority of articles published about alginate-based films, this method was used. However, solvent casting methods cannot be scaled up to an industrial level [27]. Difficulties in thickness control, limited surface area, low productivity, and long water evaporation time (2–3 days) are problems associated with this method. In addition, the production of multiphase systems such as blends with other polymers or reinforced composites is rarely achieved [47,77].

On the other hand, hot-compounding methods are capable of addressing most of these problems and can be used as an effective method for producing alginate-based film in industry. In this method, the material combines under heat and shearing force by extrusion or calendaring, and hot-press molding, injection molding, or film blowing [35]. Despite being two-step techniques, hot-compounding methods are preferable to the conventional solvent casting method since the evaporation step is not needed, which saves time and energy. It offers more controllability of the thickness, and the addition of plasticizers, reinforcing agents, and other polymers in the internal mixer is more convenient. These methods are generally used for processing plastic materials in industry. Thermo-mechanical mixing methods were developed for the first time in the 1990s for processing starch as a polysaccharide [113] and later were applied to producing chitosan-based films [85]. Plasticizers, such as water and polyols as processing aids, play a prominent role in the feasibility of this method because they decrease the processing temperature and prevent premature degradation of polysaccharides before reaching the molten state. In addition, the softening effect of plasticizers can improve the flow properties of the blend, with reduced viscosity and better dispersion of fillers in the matrix [35].

In 2017, Gao et al. [19,47], produced plasticized alginate-based films by using a thermo-mechanical mixing method and found that this method is an efficient technique for producing alginate-based film on a large scale. They also used water and polyols as a destructuring agent. The extruded material was homogeneous, and the compression-molded film was soft and transparent. However, there were traces of remnant alginate in the film, indicating a poor destructuring of the alginate powder. As a result of these defects, the films produced by the thermo-mechanical mixing method possessed inferior mechanical properties than the films produced by solvent casting. Accordingly, more studies are needed in the future to improve the quality and mechanical properties of these films through a better understanding of the microstructure, segregation, and phase separation of alginate-based mixtures.

Despite all the advantages of thermo-mechanical mixing methods, there are still some difficulties that should be considered. First of all, the existing machines and infrastructure related to hot-compounding are generally suitable for synthetic plastics, not biopolymers [27]. Unlike synthetic plastics that experience melting while processing, alginate-based mixtures may experience gelatinization as a result of adding crosslinking agents during the processing step, so different kinds of machines and equipment should be designed to meet those requirements. On the other hand, the drying process of biopolymers is completely different. Accordingly, special attention is needed to adapt machinery to meet the specific requirements of biopolymers. Secondly, most of the hot-compounding methods recommended in scientific studies concern the batches producing the films. Munhoz et al. [114] proposed a continuous edible film fabrication by spreading a wet layer of film-forming solution on a conveyor and hot-air drying. They produced 0.03 m^2^ of film per minute, and the total drying time was 7 min. Similar studies on the continuous film production of different biopolymers should be performed to investigate the effect of this technique on mechanical properties. Lastly, special attention should be paid to process and post-extrusion parameters, such as pressure at the die, barrel temperature, screw speed, die diameter, hot-press pressure and temperature, and blowing and injection conditions since the properties of films are significantly influenced by these parameters. Moreover, further investigations are required to optimize the parameters and understand the mechanisms of plasticizers and film formation, and thereby enable the successful industrial production of alginate-based films.

### 7.2. Lack of Knowledge

Although there are numerous articles about the effects of different kinds of plasticizers on the thermal and mechanical properties, and water vapor permeability of alginate-based films, still there are some aspects that have not received sufficient attention.

First of all, there is very limited research work on the structure–properties relationship of the alginate–plasticizer system [15]. Fundamental research on the degree of crystallinity, free volume, and glass transition temperature of the alginate films before and after adding plasticizer are vitally important to better understand the plasticization mechanisms and thereby select the most suitable plasticizer type as well as its concentration. Secondly, articles on the processing and hot-compounding of alginate are very rare [47]. For mass-producing alginate-based films for industrial applications, adequate knowledge about these techniques is highly important; hence, special attention should be given in future research works to optimizing the process conditions and plasticizer content for the batch or continuous fabrication of alginate-based films.

Moreover, alginate is a hydrophilic biopolymer and possesses high water vapor permeability. Hydrophobic plasticizers such as oils are capable of decreasing the WVP of the alginate films considerably, making the films more hydrophobic and comparable with their synthetic counterparts [51]. However, there has not been sufficient research work on the effects of these kinds of plasticizers on alginate films.

Furthermore, most of the studies on the barrier properties of plasticized alginate concern only the WVP of the films, and there are very limited works on other barrier properties, such as oil permeability, related to the microstructure of alginate-based films [15]. This is especially important when the films are used as food packaging, where oily ingredients can affect the ability of the film to preserve the food.

For most applications, the solubility of alginate-based films is usually reduced by the incorporation of crosslinking agents. It is well known that plasticizers and crosslinking agents may have some synergistic effect, and understanding their relationship is very important in designing an appropriate film for a specific application [8,13,62].

Moreover, knowledge about the various mechanical properties is very important to ensure the performance of the films in different applications is satisfactory. While the tensile properties of alginate-based films (YM, TS, and EB) have been studied to a large extent, there is a serious lack of knowledge about other mechanical properties, such as puncture strength, tear resistance, seal strength, and scratch resistance [27,102].

Lastly, alginate and other biopolymers usually experience aging during storage and this may bring about many problems in handling and usability of the films. To increase the efficiency of the films, it is vitally important that the properties of the films be sufficiently stable [115]. In addition, the secondary bonds between alginate and the plasticizer, as well as the low molecular weight of the plasticizer, may lead to the migration of plasticizers to the surface, which makes the film brittle. However, to the best of our knowledge, there are few studies in the literature concerning the aging of alginate and the impact of the migration of plasticizers [21].

## 8. Conclusions

In this review paper, the effects of different plasticizers on the microstructure of alginate-based films were studied. The major findings and results are summarized as follows:Pure alginate films are brittle with poor mechanical properties, and compared with synthetic plastics, possess inferior properties: hence, extra research effort is needed to improve their properties and flexibility to levels comparable with their synthetic counterparts.Among the plasticizers used for plasticizing alginate films, glycerol has proved to be the most studied because of its low cost and efficiency. However, glycerol is a very small molecule, highly hygroscopic, and soluble in water. These characteristics limit the use of glycerol-plasticized films in humid environments because of leaching, migration, and increasing the WVP.Hydrophobic plasticizers, such as oils, fatty acids, and citric acids, have the potential to decrease the hygroscopic characteristics of alginate films and reduce the WVP to make films comparable with synthetic plastics. Unfortunately, there have only been a few studies on this plasticizer.Segregation phenomenon (phase separation), anti-plasticization effects, leaching, migration, and evaporation from the surface are the unexpected problems associated with plasticizers. To avoid the segregation phenomenon and anti-plasticization effects, an optimal amount of plasticizer should be added to the polymer. Leaching, migration, and evaporation from the surface can be avoided by applying an effective plasticization process.Most of the alginate films are produced by a solvent casting method at a laboratory scale. This method is unsuitable for scaling up to an industrial scale because of the difficulties in thickness control, blending, long drying time, and high cost. Hence, special attention should be paid to this field in future works.

Based on these observations, it can be concluded that more research work is needed on the plasticity of alginate to address the problems and difficulties discussed and further develop this biopolymer for different applications.

## Figures and Tables

**Figure 1 molecules-28-06637-f001:**
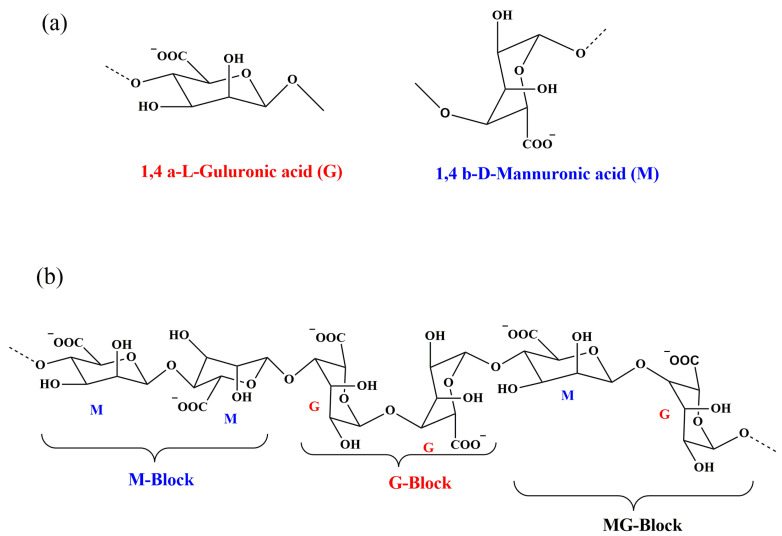
Alginate structure: (**a**) arrangement of the molecules in alginate; (**b**) three regions of M-blocks, G-blocks, and alternating sequences of MG-blocks in alginate structure (after [14]).

**Figure 2 molecules-28-06637-f002:**
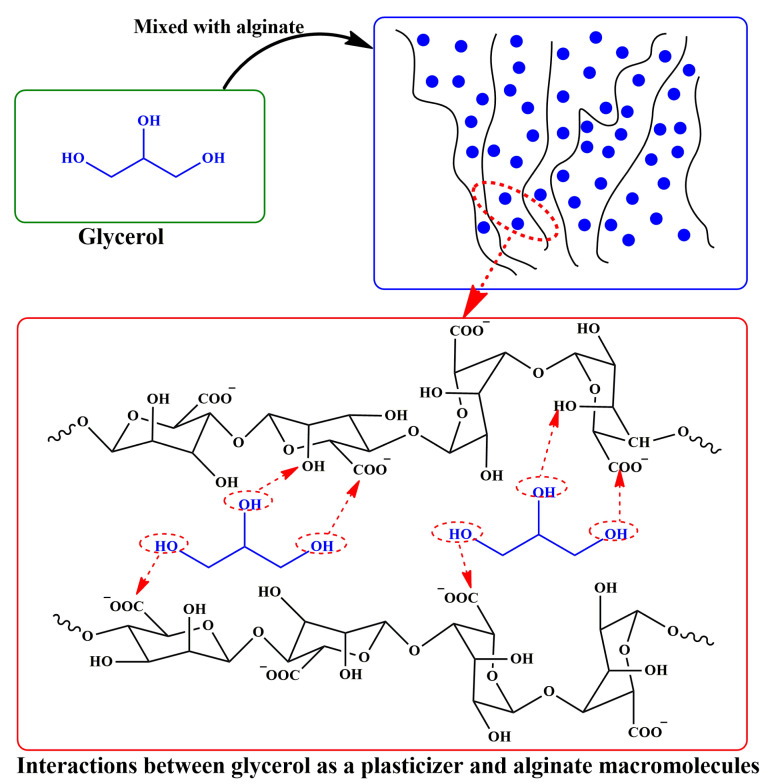
Schematic illustration of the effect of the most common plasticizer, glycerol, on alginate chains (The color scheme is used for the sake of clarity).

**Figure 3 molecules-28-06637-f003:**
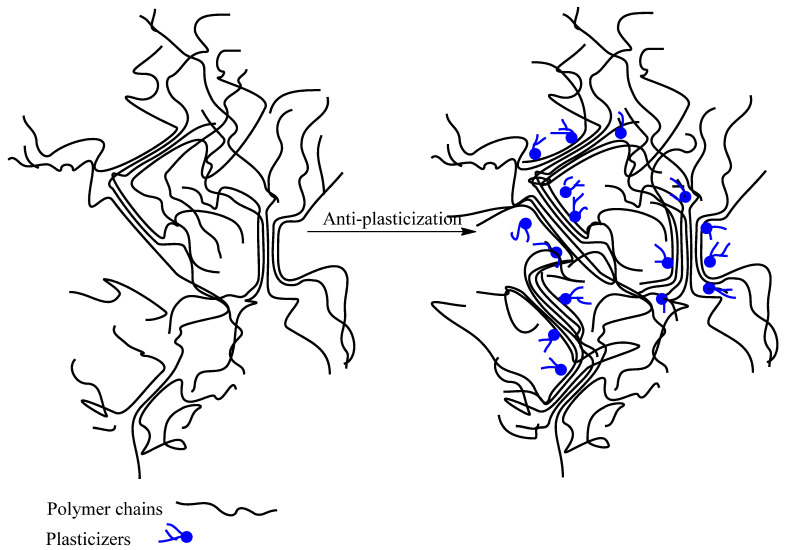
Mechanism of anti-plasticization action (after [38]).

**Figure 4 molecules-28-06637-f004:**
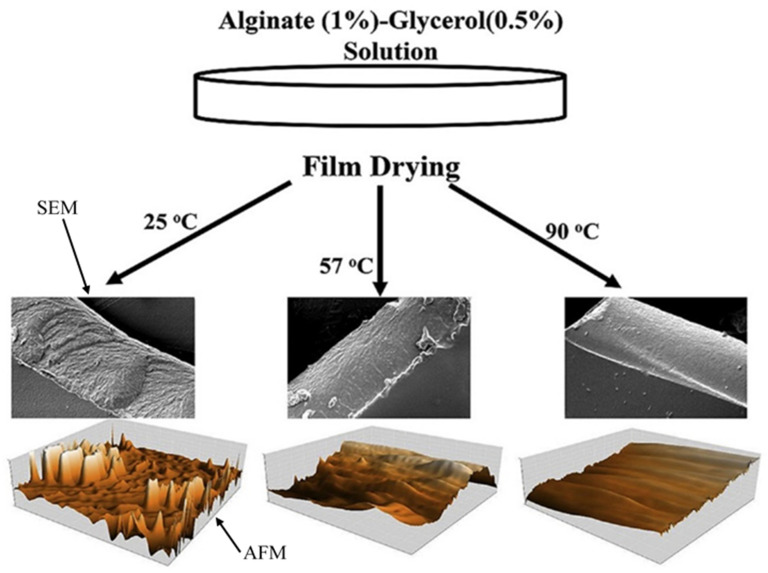
SEM and AFM results of alginate films dried at 25, 57, and 90 °C [77].

**Table 1 molecules-28-06637-t001:** Plasticizers that are used in alginate-based films.

Bio-Polymer	Plasticizer	Findings and Results	Refs.
Alginate and gelatin	Glycerol and water	Increasing RH led to increased TS and decreased EB, and glycerol increased the flexibility of the films without altering TS.	[52]
Alginate and pectin	50% glycerol and water (53% RH)	The absorbed water during conditioning had a plasticizing effect on the films.	[53]
Chitosan and alginate	25, 40, and 50% glycerol and sorbitol–water	A dramatic decrease in T_g_ with the incorporation of glycerol and water.	[54]
Alginate (M/G = 0.45 and M/G = 1.5)	Fructose, glycerol, sorbitol, and polyethylene glycol (PEG-8000)	WVP was higher for the films conditioned at higher RH; PEG-plasticized films were opaque because of phase separation.	[31]
Alginate	Glycerol and water	Films conditioned at 100% RH had higher EB and lower TS and YM than films conditioned at 57% RH.	[55]
Alginate and other carbohydrates	water	Wet alginate films had higher EB than dried films because of the plasticizing effect of water.	[56]
Alginate	Glycerol and water	There was a remarkable decrease in the degradation temperature of the film plasticized with 50% glycerol. Increasing glycerol content beyond 30% led to the segregation phenomenon.	[19]
Alginate	Glycerol, sorbitol, and water	The glycerol-plasticized film had lower T_g_ than the sorbitol-plasticized one, The films obtained from the solvent casting method had higher TS and YM, but lower EB compared with the thermo-mechanical mixing method.	[47]
Corn starch (CS) and sodium alginate (SA)	15% glycerol and water	A twin-screw extruder was used for blending the materials, and glycerol and water decreased the processing temperature.	[57]
Alginate	Glycerol (20–40%) and sorbitol (30–50%)	Although glycerol is a more effective plasticizer based on mass content, the plasticizing efficiency of sorbitol was higher at the molecular basis.	[15]
Alginate	Glycerol	Glycerol and calcium chloride (crosslinker) had a synergistic effect on the mechanical properties of the films, and beyond a certain limit they had a deteriorating effect.	[13]
Alginate (high guluronic acid (Ap) and low guluronic acid (Ar))	Glycerol	Ap polymer was effectively plasticized because of its buckled structure.	[17]
Alginate and low acyl gellan	Glycerol	The optimal concentration of glycerol was 8% *v*/*v*.	[58]
Alginate	Glycerol	At temperatures above 40 °C, a significant amount of glycerol was lost.	[59]
Alginate/pectin	Glycerol	When the plasticizer concentration was above a critical limit, phase separation could be observed on the surface of the film.	[29]
Alginate	Polyglycerol	Polyglycerol had an anti-plasticization effect on alginate because of the presence of high amounts of hydroxyl groups in polyglycerol.	[45]
Alginate	glycerin and polyethylene glycol 400 (PEG400)	Glycerin was a better plasticizer than PEG400 and gave more flexibility to the films because of the lower molecular weight of glycerin.	[30]
Alginate	glycerol (GLY), diethylene glycol (DEG), and polyethylene glycol (PEG)	WVP of films plasticized with PEG and DEG was lower than that of GLY-plasticized film.	[42]
Alginate	Glycerol, Xylitol, and mannitol	Glycerol and xylitol-plasticized films were more transparent and uniform than the mannitol-plasticized film, but they had higher WVP.	[60]
Alginate and vegetable oils	Glycerol and sorbitol (0–20%)	The surface tension did not alter by the addition of the plasticizers, but vegetable oils diminished the surface tension.	[61]
Pullulan and alginate	glycerol, sorbitol, xylitol, and fructose	Sorbitol- and fructose-plasticized blend films exhibited the lowest and similar EB at any given plasticizer concentrations compared with glycerol and xylitol-plasticized films, with the fructose-plasticized film being even more brittle with higher TS and lower EB.	[36]
Alginate and pectin	Glycerol	Increasing glycerol content promoted the WVP of the films.	[62]
Alginate and pectin	33% Polyglycerol	Higher swelling degree cross-linked film with the addition of polyglycerol.	[8]
Alginate and apple puree	Glycerol, rapeseed oil, coconut oil, hazelnut oil, and sugars in the apple puree	The T_g_ decreased with the addition of vegetable oils and apple puree, so they had a plasticizing effect.	[63]
Alginate	Glycerol and oregano essential oil (OEO)	Higher EB and lower WVP and TS observed with the incorporation of OEO.	[51]
Alginate	Glycerol, castor oil (CO)	The incorporation of CO led to increased EB and decreased TS and WVP.	[64]
Alginate	Glycerol and soybean oil	At high calcium chloride concentrations, the EB of alginate decreased with increasing oil concentrations. WVP decreased with the addition of oil.	[65]
Alginate (2–6%) and virgin coconut oil	Glycerol (10%)	To decrease the surface tension of oil and alginate, ethanol was used.	[66]
Alginate/gelatin	Glycerol and canola oil	Higher EB and lower WVP and TS observed with the incorporation of canola oil.	[67]
Alginate	Glycerol and cinnamon essential oil (CEO)	The incorporation of higher amounts of CEOs led to a decreased EB.	[68]
Alginate	Glycerol and cinnamon essential oil (CEO)	The incorporation of the CEO led to an increased EB and WVP and decreased TS.	[69]
Alginate/garlic oil	-	Garlic oil increased EB and decreased TS of the film, and WVP increased remarkably with increasing oil content.	[70]
Alginate	Glycerol, essential oils (Eos)	Oil droplets had a plasticizing effect by decreasing interactions between chains.	[71]
Alginate/apple puree	Glycerol and plant essential oils	EB increased with the addition of the oil, but TS decreased.	[72]
Soy protein isolate/alginate	Stearic acid and lauric acid	TS and EB of the films decreased with the incorporation of the fatty acids; however, EB increased at higher concentrations of auric acid. WVP value decreased at lower amounts of fatty acids, but it increased at higher amounts.	[50]
Alginate	Glycerol and oleic acid	Oleic acid behaved like a second plasticizer.	[3]
Alginate	Glycerol, tri-butyl citrate (TC)	TC-plasticized films were opaque; T_g_ and TS increased with the addition of TC; EB decreased with the addition of TC.	[33]
Alginate	Citric acid (CA)	CA at higher concentrations had a plasticizing effect.	[34]
Chitosan/alginate	Triacetin, glycerol, and Ionic liquid	Triacetin-plasticized films were brittle and thermally stable.	[44]
Alginate	Graft copolymerization of itaconic acid (internal plasticization)	The T_g_ value of the grafted alginate film was lower, indicating the plasticizing effect of itaconic acid.	[73]
Natural polysaccharides such as alginate	Graft copolymerization of polyacrylonitrile (internal plasticization)	The grafted chains might act as internal plasticizers because of the reduced Tg.	[74]
Alginate	Lemon and fennel wastes (contain pectin-like polymers)	T_g_ and degradation temperature decreased, but the EB and TS of the films increased with the incorporation of the plasticizers.	[75]
Alginate/lignin	Glycerol and lignin	Lignin exerts an apparent plasticizing effect on alginate by reducing the intermolecular interaction between chains and decreasing the tensile strength of the films.	[76]
Gluronate-rich (MG) and mannuronate-rich (MC) alginate	Water and hot air	Plasticity was decreased by increasing the drying temperature to 60 °C, Hot air at 80 °C induced plasticity because of the formation of bubbles and degradation of alginate molecules.	[22]
Alginate	Glycerol	The amount of glycerol in the dried films was decreased by increasing the drying temperature; hence, the properties of the film were affected.	[77]
Alginate	-	According to thermo-mechanical analysis, the films prepared at 80 °C were more plasticized than the films produced at lower temperatures.	[78]

RH = Relative Humidity, YM = Young’s modulus, TS = Tensile Strength, EB = Elongation at the break, WVP = Water Vapor Permeability, T_g_ = Glass Transition Temperature.

## Data Availability

Not applicable.

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
