# Peer review of "A Review of the Effect of Plasticizers on the Physical and Mechanical Properties of Alginate-Based Films"

_molecules, 2023, doi:10.3390/molecules28186637_

Round 1
Reviewer 1 Report
The manuscript entitled " A review of the effect of plasticizers on the physical and mechanical properties of alginate-based films” describes the effect of different types of plasticizers on water vapor permeability, and thermal, mechanical, and physical properties of alginate-based films. It presents scientific relevance for the area of Food industry and chemical industry.
After consulting www.sciencedirect.com and https://pubmed.ncbi.nlm.nih.gov/, publications were found for some authors involving the theme. However, you need to change some details:
-line 115 it is not the same font
-line 117 Which of the possible applications receives in the article? Cosmetic, food industry
medical applications?
-line in figure 2, review the size
-table 1 to revise the abbreviations to be uniform
The article is very rich in split information (such as Polyols)
Congratulations, an interesting research article. Good results, with pertinent explications.
Reviewer 2 Report
1. Page no: 3 last three lines are in different font
2. Why the above 40% of glycerol will lower the elongation?
3. How to Taylor the glass transition temperature in order to use the natural extract as a plasticizer?
4. What is the glass transition temperature of castor oil?
5. Can we use the plasticizer as a crosslinker?
6. As you said oils will decrease the TS and increase the EB of the film. Then how cinnamon oil will decrease the EB. Give your hypothesis.
7. How to encounter the segregation phenomena when adding the plasticizer?
8. Page no 24 in subtitle 6.4 last para change the font
9. What is the ideal polymer / solvent ratio?
10. What is the alternating best method than the casting method?
11. What about the storage stability of the plasticized alginate films?
OK
